# Seasonal and Interhemispheric Effects on the Diurnal Evolution of EIA: Assessed by IGS TEC and IRI-2016 over Peruvian and Indian Sectors

Xin Wan [1,2], Jiahao Zhong [1,2,*], Chao Xiong [3], Hui Wang [3], Yiwen Liu [4], Qiaoling Li [1,2,5], Jiawei Kuai [6] and Jun Cui [1]

1   Planetary Environmental and Astrobiological Research Laboratory (PEARL), School of Atmospheric Sciences, Sun Yat-sen University, Zhuhai 519082, China; wanx7@mail.sysu.edu.cn (X.W.); liqling6@mail.sysu.edu.cn (Q.L.); cuijun7@mail.sysu.edu.cn (J.C.)
2   Key Laboratory of Tropical Atmosphere-Ocean System, Ministry of Education, Zhuhai 519082, China
3   Department of Space Physics, College of Electronic Information, Wuhan University, Wuhan 430079, China; xiongchao@whu.edu.cn (C.X.); h.wang@whu.edu.cn (H.W.)
4   School of Physics and Electronic Information, Shangrao Normal University, Shangrao 334000, China; liuyiwen@whu.edu.cn
5   Mengcheng National Geophysical Observatory, University of Science and Technology of China, Hefei 230026, China
6   College of Astronautics, Nanjing University of Aeronautics and Astronautics, Nanjing 210016, China; jwkuai@nuaa.edu.cn
*   Correspondence: zhongjh55@mail.sysu.edu.cn

**Abstract:** The global total electron content (TEC) map in 2013, retrieved from the International Global Navigation Satellite Systems (GNSS) Service (IGS), and the International Reference Ionosphere (IRI-2016) model are used to monitor the diurnal evolution of the equatorial ionization anomaly (EIA). The statistics are conducted during geomagnetic quiet periods in the Peruvian and Indian sectors, where the equatorial electrojet (EEJ) data and reliable TEC are available. The EEJ is used as a proxy to determine whether the EIA structure is fully developed. Most of the previous studies focused on the period in which the EIA is well developed, while the period before EIA emergence is usually neglected. To characterize dynamics accounting for the full development of EIA, we defined and statistically analyzed the onset, first emergence, and the peaks of the northern crest and southern crest based on the proposed crest-to-trough difference (CTD) profiles. These time points extracted from IGS TEC show typical annual cycles in the Indian sector which can be summarized as winter hemispheric priority, i.e., the development of EIA in the winter hemisphere is ahead of that in the summer hemisphere. However, these same time points show abnormal semiannual cycles in the Peruvian sector, that is, EIA develops earlier during two equinoxes/solstices in the northern/southern hemisphere. We suggest that the onset of EIA is a consequence of the equilibrium between sunlight ionization and ambipolar diffusion. However, the latter term is not considered in modeling the topside ionosphere in IRI-2016, which results in a poor capacity in IRI to describe the diurnal evolution of EIA. Meridional neutral wind's modulation on the ambipolar diffusion can explain the annual cycle observed in the Indian sector, while the semiannual variation seen in the Peruvian sector might be due to additional competing effects induced by the F region height changes.

**Keywords:** GPS TEC; EIA; diurnal evolution; seasonal variation

## 1. Introduction

The equatorial ionization anomaly (EIA) is the result of ionization and electrodynamic processes in the ionosphere, representing a plasma density trough at the equator and two crests approximately 15° to the north and south. The formation process of EIA can be characterized as a fountain effect. That is, the daytime E region wind dynamo produces an

eastward electric field to move the plasma upward via E×B force; the increasing gravitational potential and pressure gradient would finally result in ambipolar diffusion, which transports the equatorial source plasma downward/poleward to the north and south, to create double plasma density crests [1–10].

Extensive studies have investigated the development of the EIA under various conditions configured by solar radiation, season, local time (LT), longitude, and geomagnetic disturbances, to reveal the impacts of those parameters on the ionizations, electrodynamics, and neutral dynamics behind the fountain effects. Yeh et al. [1] found that EIA crests generally begin to appear at 0900 LT, and the farthest latitude of the crests during daytime is correlated with the level of the fountain effect and the local ionospheric total electron content (TEC). Liu et al. [2] reported that the crest-to-trough ratio (CTR) of EIAs observed by the CHAMP satellite gradually increases from morning to noon, reaching its maximum value between 1800 and 1900 LT as a result of pre-reversal enhancement (PRE). Xiong et al. [3] reported that the electron density and magnetic latitudes of both EIA crests peak at approximately 1400 LT. The local time variation of the electron density crest during daylight hours is similar to that of the trough but with a 1–2 h delay [4].

Another interesting aspect of the EIA is the interhemispheric asymmetry, which is most pronounced during solstitial seasons. In the morning hours, the crest in the winter hemisphere generally forms earlier and has a greater magnitude than that in the summer hemisphere; during the afternoon, the summer hemisphere features the larger EIA crest [3,6–10]. The transition time of this interhemispheric EIA asymmetry occurs at around 1200–1400 LT, depending on solar activity levels [6,7]. In addition, this transition time is a function of observed altitude, since the time lag of fountain effects varies with height [3]. Interhemispheric EIA asymmetry has also been reported during equinoxes [7,10,11]. Neutral wind variations associated with displacements of the geographic and geomagnetic equators as well as magnetic declination angles have been proposed as the main factors affecting EIA asymmetry [10]. Balan et al. [11] argued that the displacement of the geographic and geomagnetic equators is a more significant factor than the declination angle.

The above-mentioned studies mainly focus on the EIA that exhibits clear double crests signature. However, before the first emergence of EIA double crests, the fountain-like processes had already been launched, but this stage receives much less attention in the research community. In detail, the sunlit ionization process is instantaneous, while the mechanical transportation process is much slower. Thus, at the beginning, stronger sunlit ionization actually would cause faster plasma accumulation at the subsolar position near the equator. The fountain effects take time to compete against the uneven ionization, to form EIA. The contributions of various physical processes during different stages of the EIA diurnal evolution are still not well known. In particular, the dynamic process before the emergence of the EIA crest is usually neglected and had not been investigated yet.

Note that the sunlit ionization and electrodynamical transportation have different reference equators, i.e., the geographic and geomagnetic equators. The displacement between the two equators varies at different longitudes, which would significantly impact the EIA evolution. Moreover, meridional thermospheric wind, which drags the ion along with it, is reported to impact the EIA evolution in two opposite ways. On the one hand, the transequatorial thermospheric wind pushes the plasma along the field line to contribute/counter the ambipolar diffusion in the winter/summer hemisphere, leading to a winter hemispheric priority during EIA's development [6,7]. On the other hand, the transequatorial thermospheric wind would lift/lower the F region height in the summer/winter hemisphere, leading to stronger/weaker intensity (i.e., TEC) of EIA crest [11–13]. It can be seen that the inclusion of thermospheric neutral wind effects could make the EIA evolution more complicated, the dominant seasonal cycles at different longitudes are still not well understood, and the physical processes and mechanisms involved are still in debate [6,7,12–14].

We dedicate this study not only to monitoring the interhemispheric asymmetry in a traditional way that focuses on the intensity of EIA, but also to trying to characterize the

detailed time evolution to clarify the dynamical competition and cooperation between the sunlit ionization, ambipolar diffusion, and neutral wind drag. The TEC maps provided by IGS [15] are used under the geomagnetic quiet condition in 2013 in the Peruvian and Indian sectors. At low geomagnetic latitudes, these two sectors are both deployed with magnetometers to retrieve an equatorial electrojet (EEJ) that can be used as a proxy to select days with developed EIA, and GNSS receivers that provide reliable TEC product to investigate the seasonal and interhemispheric effects on the diurnal evolution of EIA. The International Reference Ionosphere (IRI-2016) model, widely recognized as a powerful tool to represent the climatological behavior of the ionosphere [16–19], is also adopted to check whether the empirical model can capture the real features of the EIA evolution.

In Section 2, we provide descriptions of the datasets. In Section 3, we introduce how the geomagnetic quiet days and days with well-developed EIAs are sorted out. Section 4 presents the seasonal/local variation of EIA, followed by detailed statistics of crests and trough variations. The crest-to-trough difference (CTD), is defined and described in Section 4.2. The EIA onset time, first emergence and peaks are further derived from CTD profiles and we presented the statistical results. In Section 5, we discuss the physical mechanisms involved. Conclusions of this study are provided in Section 6.

## 2. Dataset

### 2.1. IGS TEC Maps and the IRI-2016 Model

The IGS TEC data is an interpolated data product based on the TEC measurements from ground-based GNSS receivers that are distributed mainly over the continents [15]. Thus, the IGS TEC should provide trustable EIA observation over Indian and Peruvian sectors. The dataset has a time resolution of 15 min and spatial resolution of $2.5° \times 5°$ in geographic latitude and longitude.

The distribution of GPS receivers over the oceanic region is much sparser than the continents, IGS TEC was found to overestimate Jason-2/3 derived TEC by more than 5 TECU [20]. This overestimation would possibly impact the climatological behavior of the retrieved EIA features. Thus, whether the IGS TEC is suitable to be extendedly applied to the longitudes over the oceanic region remains unclear. We choose another candidate data source from the IRI-2016 model, a widely used empirical ionospheric model, and it was recently improved with a new hmF2 model based on a new database from the worldwide network of ionosondes [17,19]. The IRI-2016 is used to retrieve the EIA feature and compare with IGS TEC data. This comparison would not only help to assess the performance of the empirical model in describing the regional EIA evolution, but also evaluate the feasibility of whether the empirical model can be applied to extend EIA study over the oceanic region.

For a given location, time, and date, the IRI model provides monthly averages of the ionospheric parameters, including electron density, electron temperature, ion temperature, ion composition, and TEC from an altitude range of 50–2000 km [17–19]. Options were set to calculate the TEC from the IRI-2016 model: The fof2 storm model was switched off, the Shubin-cosmic option was used for the hmf2 model, and NeQuick was used as the topside model. The maximum height of the TEC calculation was set to 2000 km.

### 2.2. The EEJ Derived from Ground-Based Magnetometers

Ground-based paired magnetometer measurements over the Peruvian and Indian sectors in 2013 were utilized to estimate the equatorial electrojets (EEJ). As a narrow current that flows in the E region above the magnetic equator, EEJ can be extracted by removing the solar quiet (Sq) current that barely shows latitudinal dependence [21]. To estimate the EEJ, we calculated the differences of the horizontal (H) component of the geomagnetic field between the paired magnetometer, as the residual horizontal magnetic field is recognized to be caused by the EEJ [22]. The measurements of Huancayo (HUA, $-12.05°$ N, $-75.33°$ E, $0.59°$ dip latitude) and Fuquene, (FUQ,$18.11°$ N, $-66.15°$ E, $17.06°$ dip latitude) are used for the Peruvian sector, and Tirunelveli (TIR, $8.7°$ N, $77.8°$ E, $0.59°$ dip latitude) versus Alibag (ABG, $18.6°$ N, $72.9°$ S, $13.67°$ dip latitude) are used for the Indian sector [19].

*2.3. Horizontal Wind Simulated by TIEGCM*

To estimate the neutral wind effects on the development of EIA, the horizontal wind simulated from the Thermosphere Ionosphere Electrodynamics General Circulation Model (TIEGCM) is adopted. The TIEGCM is a first principle and physics-based model driven by a high-latitude electric field [23], solar EUV, and UV spectral fluxes parameterized by the F10.7 index [24].

## 3. Methodology and Methods

*3.1. Sorting Geomagnetic Quiet Days Using Kp Index*

*Kp* index is a quasi-logarithmic index, ranging in steps of 1/3 from 0 to 9, to quantify the level of the geomagnetic disturbance on a global scale [25,26]. All the data used in this study were firstly sorted under geomagnetic quiet conditions when the daily mean *Kp* values were less than 3.

*3.2. Sorting Developed EIA Using EEJ as a Proxy*

Note that sometimes the EIA structure is dismissed, which could not be used to monitor the time evolution of EIA. Thus, we firstly determine whether the EIA appears for further statistics. As mentioned previously, the formation of EIA is a product of the eastward zonal electric field; thus, the intensity of the zonal electric field could be a good indicator for the EIA development. However, the direct measurement of the electric field is rare; an alternate option is EEJ, which serves as a proxy to quantify the daytime zonal electric field. The EEJ refers to a narrow band of intense electric current flowing above the equatorial dip in the daytime E-region driven by the E-region electric field and conductivity. The EEJ mainly flows eastward corresponding to an eastward electric field, the infrequent westward flow of EEJ is called counter electrojet (CEJ) corresponding to a westward electric field. The EEJ is considered to be a suitable proxy for EIA intensity. Stolle et al. [27] found correlation coefficients greater than 0.8 between EEJ strength and EIA intensity. Venkatesh et al. [28] discovered that the daily summed EEJ strength had correlations of 0.62 and 0.72 with the EIA crest amplitude and latitude, respectively.

Figure 1 shows examples of a weak EEJ profile (Figure 1a) and a CEJ profile (Figure 1b), with associated TEC maps for the day. The double-crest structure barely formed as the EEJ intensity was relatively weak (Figure 1a), while the CEJ resulted in a single peak at the equator, meaning that the EIA was inhibited completely.

It is reported that the strength of the EIA shows a better correlation with integrated EEJ values than the daily maximum of EEJ [28]. In addition, the time delay of the EIA response to EEJ strength is 2–3 h [27,28]. To identify the days with weak EEJ or CEJ, the averaged ΔH at an LT bin of 0800–1200 is calculated first. Please note that the Weak EEJ or CEJ were combined to be referred to as WEC in this study. Days with WEC were then determined when the averaged ΔH components were less than the threshold value; the remainder of the quiet days were recognized as EEJ days. Our experiments (not shown here) showed that the EEJ intensity was generally higher in the Peruvian sector than in the Indian sector, probably as a result of tidal effects. Thus, we chose two thresholds, ~70 nT and ~20 nT of averaged EEJ intensity during 0800–1200 LT for the Peruvian and Indian sectors, respectively.

Figure 2 shows the day numbers of the WEC and EEJ cases at longitudes of $-75°$ E and 75° E, which represent the Peruvian and Indian sectors, respectively. The WEC case showed a preferential occurrence during the two solstices at both longitudes, consistent with previous studies which had found that the EEJ is characterized by solstitial minima [29–31]. However, for a given longitude, the WEC showed specific seasonal preferences as well. There were more WEC days around the December Solstice (Nov, Dec, Jan, Feb) than the June Solstice (May, Jun, Jul, Aug) in the Peruvian sector, with this preference being reversed in the Indian sector. Considering the northward–southward deviation of the dip equator from the geographic equator at both the Indian and Peruvian longitudes, this implied that

more WEC occurred when the dip equator is in the summer hemisphere. One may also note that the WEC day is missed in Mar, Apr, Aug, Sep, and Oct in the Indian sector (Figure 2b).

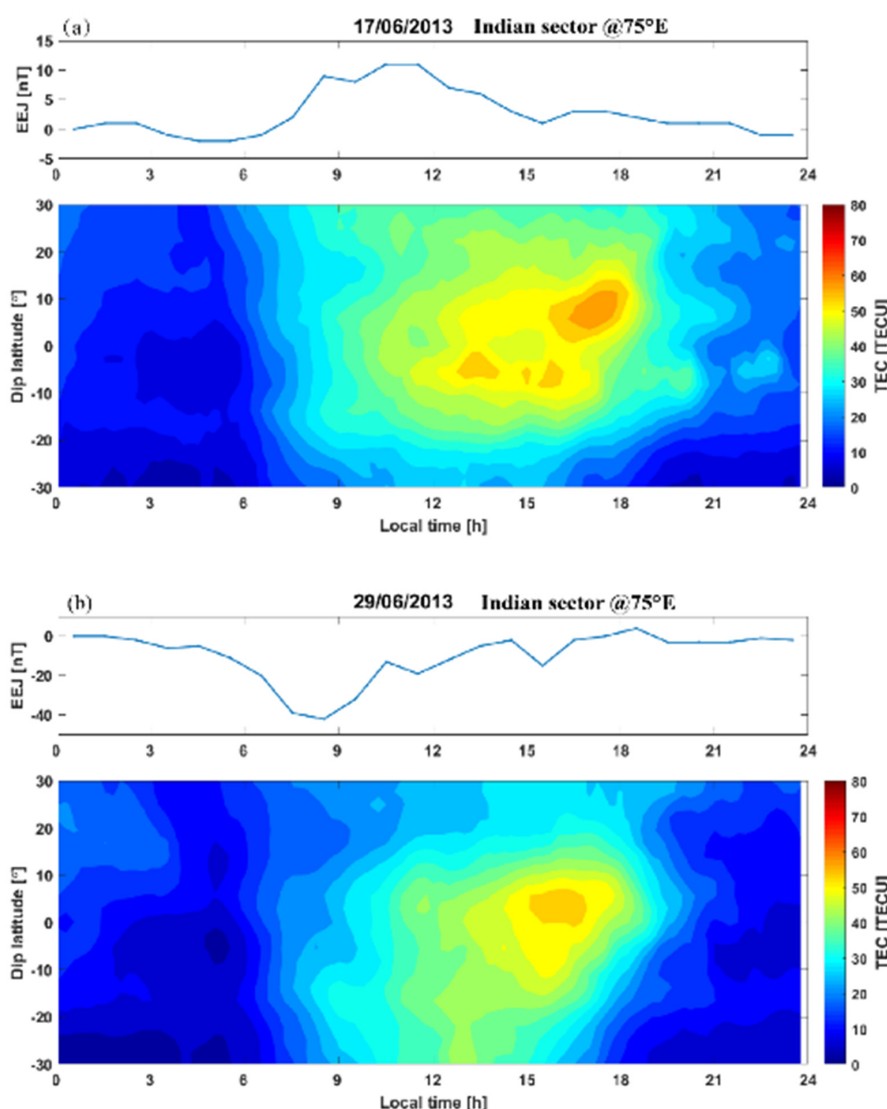

**Figure 1.** Two examples of the GPS-measured TEC map at Indian sector during (**a**) a weak EEJ day and (**b**) a CEJ day.

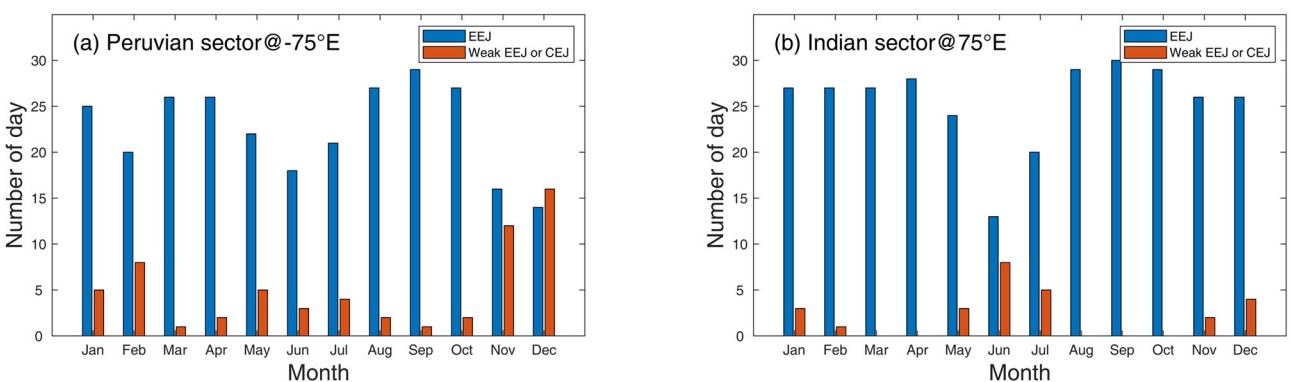

**Figure 2.** Number of days with EEJ signatures and weak EEJ or CEJ signatures at (**a**) Peruvian sector and (**b**) Indian sector.

## 4. Results

### 4.1. Overview of EIA during EEJ and Weak EEJ/CEJ Days

Figure 3 shows the TEC maps during EEJ days in the Peruvian and Indian sectors. The data shown in the left, middle, and right columns represent different seasons: December Solstice (Dec. S., which includes Nov, Dec, Jan, and Feb), June Solstice (June. S., which includes May, Jun, Jul, and Aug), and equinoxes (March, April, September, October), respectively. The daily EEJ was first plotted in Figure 3a,d; solid black lines represent four-month averages.

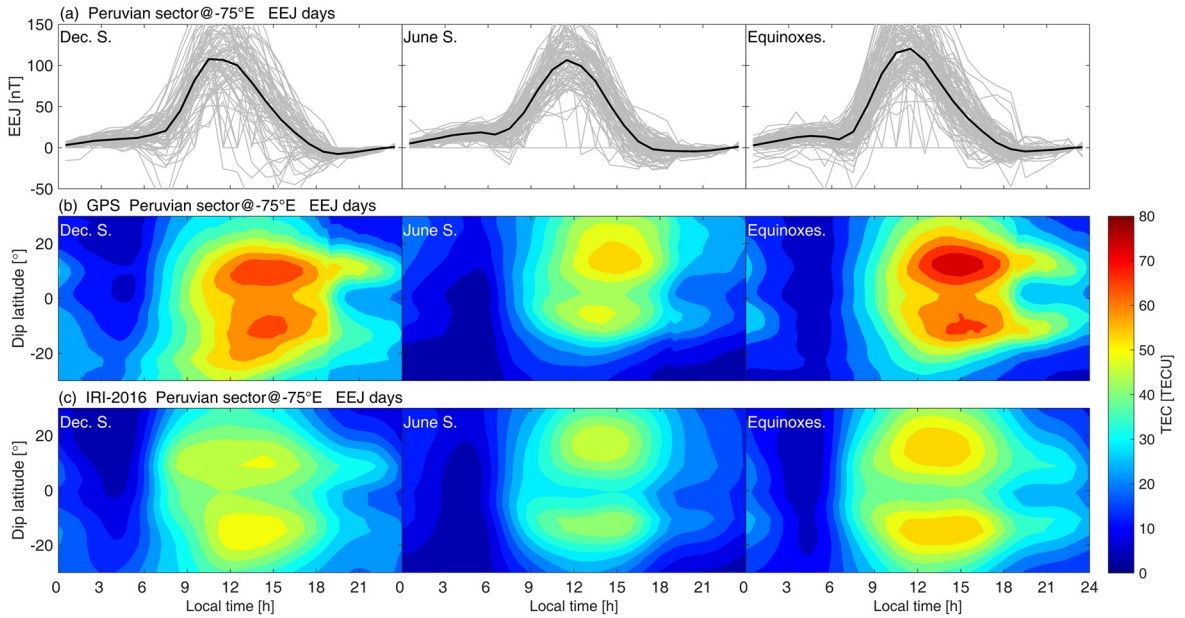

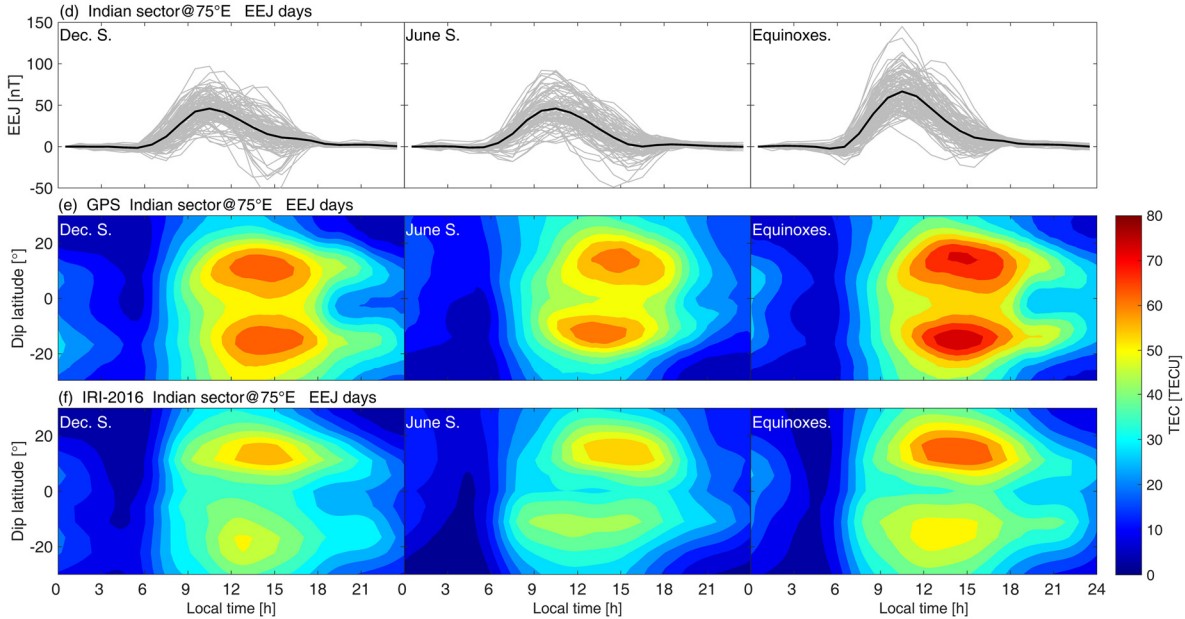

**Figure 3.** Seasonal averaged TEC maps over the Peruvian and Indian sectors during EEJ days when the averaged EEJ intensity at 0800–1200 UT was relatively high during three seasons of December Solstice (left column), June Solstice (middle column) and equinoxes (right column). (**a**) EEJ profiles of Peruvian sector; (**b**) GPS TEC at Peruvian sector; (**c**) IRI-2016 derived TEC at Peruvian sector; (**d**–**f**) are organized in a same format as (**a**–**c**) but for the Indian sector.

For GPS TEC observations at both Peruvian and Indian sectors, the most prominent feature of EIA is the interhemispheric asymmetry. During two solstices, the crests in the winter/summer hemisphere are stronger during morning/afternoon hours. Moreover, the stronger crest generally resides at a higher latitude. An exception is that at the Peruvian during Dec. S., the southern crest (in the summer hemisphere) was stronger first, which lasted an interval of 0800–0900 LT; afterward, the northern crest (in the winter hemisphere) turned out to be dominant. During equinoxes, interhemispheric asymmetry still exists in that a stronger northern/southern appeared in the Peruvian/Indian sector. Similar seasonal and longitudinal variations of the interhemispheric asymmetry have been investigated by many studies. Those studies generally accepted that the displacements of the geographic and geomagnetic equators, the different geomagnetic declinations, and the meridional neutral wind show clear seasonal and longitudinal variations, which affects the diurnal evolution of EIA [3,6–9,11].

The developed EIA signatures can also be characterized by the IRI-2016, nevertheless, with a lower absolute TEC level by approximately 20 TECU, compared with the GPS TEC. In the Peruvian sector, IRI-2016 performed stronger crests in the summer hemisphere. However, in the Indian sector, the northern crest is persistently stronger than the southern crests. Hence, it can be noticed that the IRI-2016 performed quite a different interhemispheric asymmetry pattern compared with that of the GPS observations, especially during equinoxes.

On WEC days (Figure 4), as expected, the EIA barely formed as observed by the GPS TEC. The Peruvian sector even showed a single-peak structure over the equator during the December Solstice and two equinoxes, indicating that the fountain effect was completely inhibited. For the developed EIA, the intensity is weaker than that during EEJ days; that is, the double crest was narrower, and the TEC values were reduced by approximately 15 TECU and 5 TECU in the Peruvian and Indian sectors, respectively. Note that during the equinoxes there are no WEC days for the Indian sector as displayed in the statistical day numbers in Figure 2b, thus the corresponding TEC data were not available (Figure 4d–f).

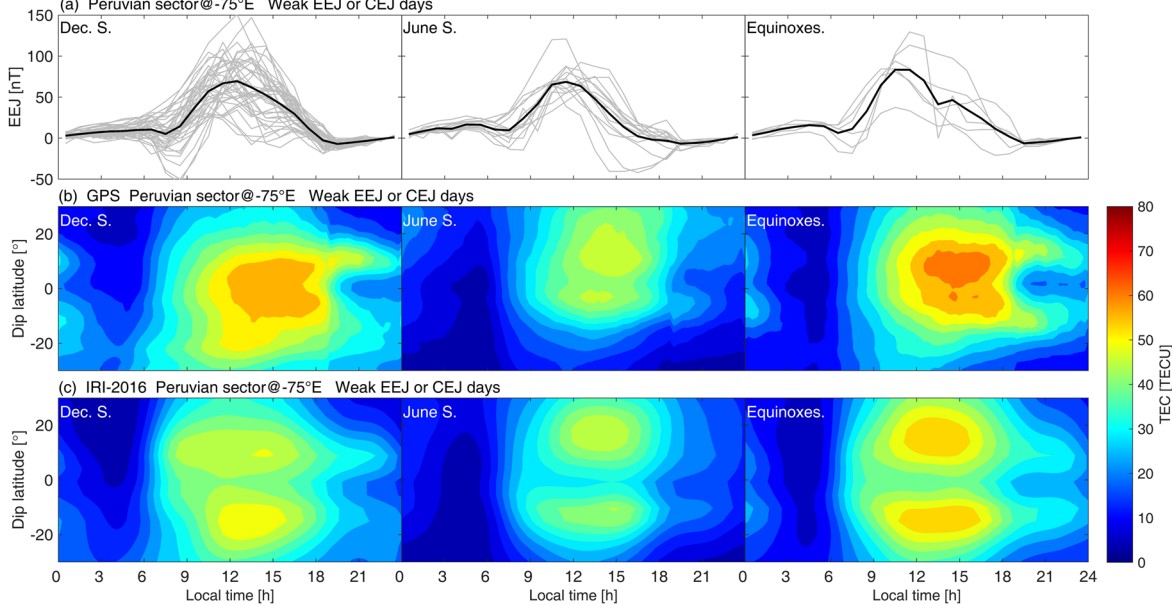

**Figure 4.** *Cont.*

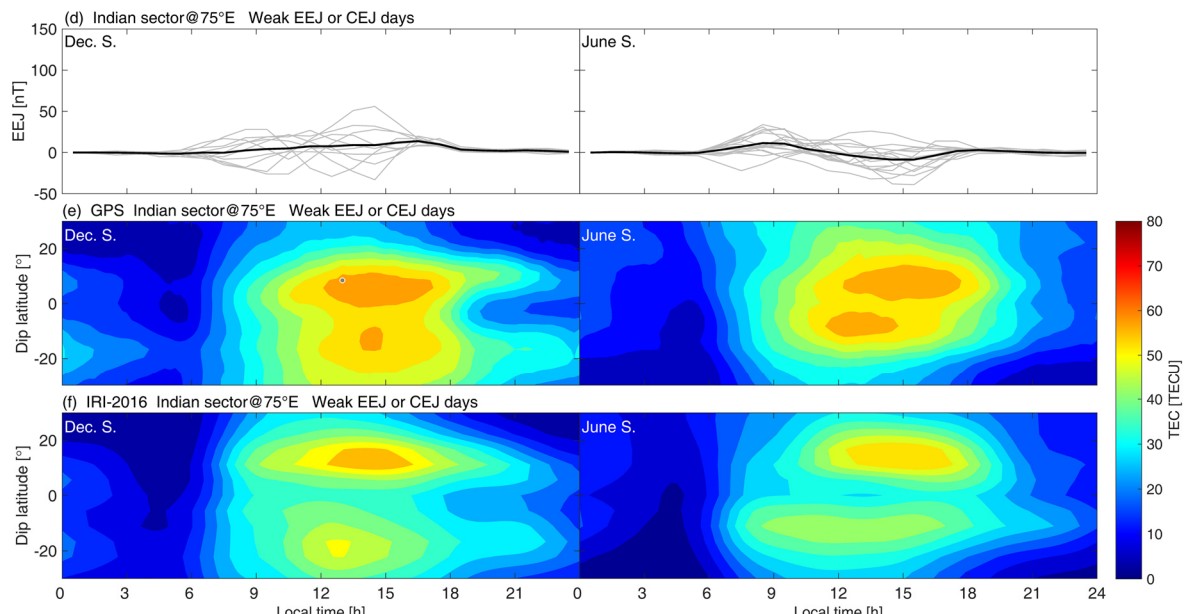

**Figure 4.** Similar to Figure 3, but during days that the average EEJ intensity at 0800–1200 UT was relatively low (refer to WEC days) at Peruvian sector (**a**–**c**) and Indian sector (**d**–**f**). The lack of equinox results in (**d**–**f**) is due to that the WEC days are not found.

However, TEC values predicted by the IRI-2016 model on WEC days remained at nearly the same levels as those during EEJ days, indicating that the model cannot capture an inhibited EIA signature. For a given time, the IRI-2016 predicted the result in terms of the monthly averaged value. Hence, this inherent characteristic of the empirical model would result in unreliable predictions for days with unusual space weather occurring during geomagnetic quiet times such as CEJs. To successfully ascertain the characteristics of daytime EIAs, we excluded those types of WEC days in the following analysis.

*4.2. Crest-to-Trough Differences (CTD)*

Despite the EIA intensity, it is easy to notice in Figures 3 and 4 that the local time of the first appearance of the crests also showed interhemispheric asymmetry. That is, a stronger crest corresponding to earlier emergence. Extensive studies have investigated the interhemispheric asymmetry of the EIA intensity; hence, in the following, we provide another perspective to investigate asymmetrical behavior concerning the time evolution of EIA. In particular, to assess the time evolution of EIA, we defined crest-to-trough difference (CTD):

$$CTD = TEC_{\text{off-equator}} - TEC_{\text{equator}} \tag{1}$$

That is, the difference between the TEC at off-equator (set as fixed dip latitude bin of 10°–15° N/S) and the TEC at the equator (set as fixed dip latitude bin of 2.5° S–2.5° N), where the EIA crest and trough normally reside, respectively. Note that the 'crest' location we defined here is not the accurate position of the EIA crest; thus, the calculation of CTD does not require the presence of a clear EIA crest feature.

Another criterion used to quantify the developed EIA intensity is the crest-to-trough ratio, defined as the ratio of the mean of the northern and southern EIA crest peak value to the minimum TEC in the EIA trough:

$$CTR = (TEC_{\text{ncrest}} + TEC_{\text{screst}})/(2 \times TEC_{\text{trough}})) \tag{2}$$

CTR is much more extensively used in other studies [2,3,27,32,33]. The CTR provides information on the overall intensity of the developed EIA normalized by the background TEC at the EIA trough. As we intend to monitor the full development of the EIA throughout

the daytime, even when the EIA crests have not been developed, the defined CTD is a more suitable parameter to achieve the goal.

Figure 5 shows an example of the northern crest-to-trough difference (NCTD) and southern crest-to-trough difference (SCTD), extracted from GPS TEC and IRI TEC maps (Figure 5a,b). GPS TEC measurements (Figure 5a) show that at 0600–1000 LT the double crest had not been developed and CTD experienced a falling and then a rising (Figure 5c). The falling occurred right after the sunrise, due to that the ionization creates more plasma around the equator. The rising should be related to the ambipolar diffusion that transports equatorial plasma to higher latitudes. Thus, the transition of the falling and rising marks the time that the transportation term dominates, i.e., the net accumulation of plasma at the off-equator exceeds that at the equator. This transition is marked as the inflection points on the CTD curved in the morning hours and is defined as the onset of EIA.

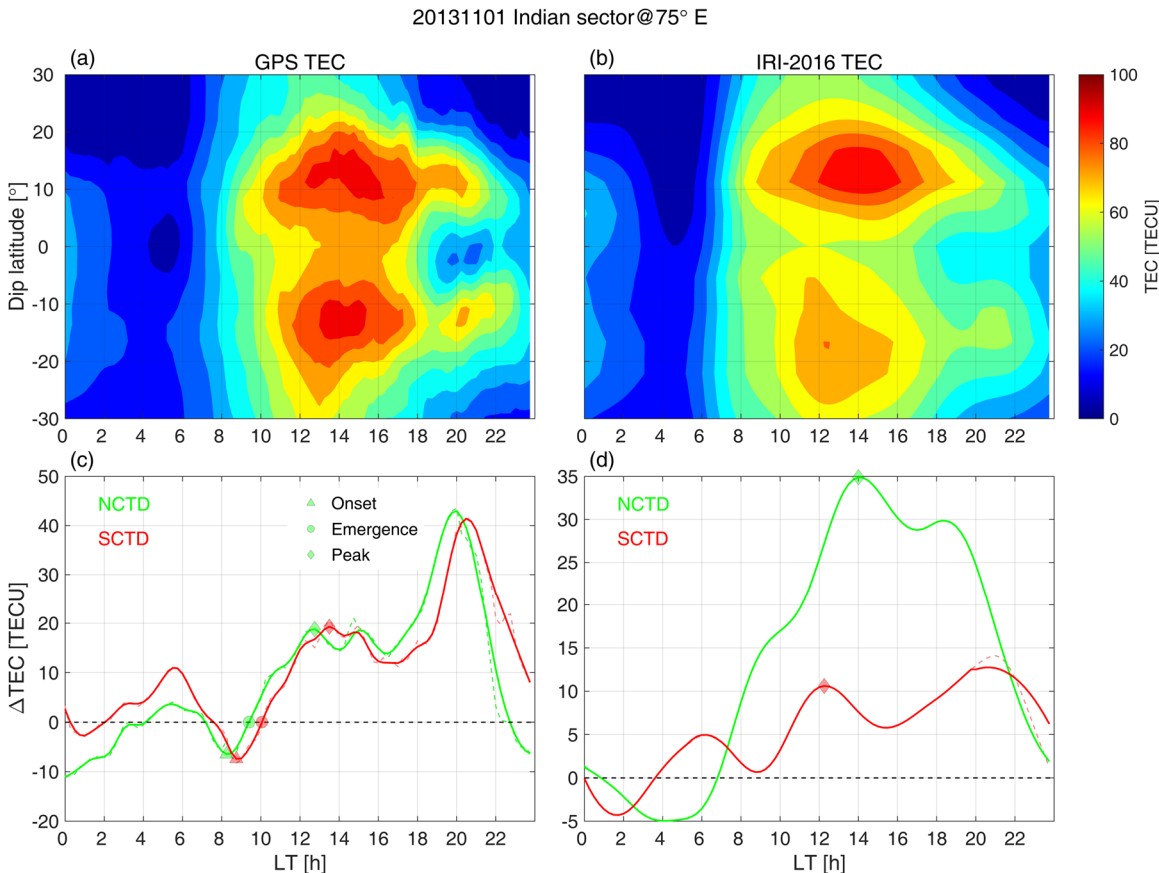

**Figure 5.** Top panels: TEC at Indian sector as a function of local time and dip latitude, using data from (**a**) IGS GPS TEC and (**b**) IRI-2016. Bottom panels: Local time evolution of the extracted NCTD and SCTD from (**c**) IGS GPS TEC and (**d**) IRI-2016. Dash curves are the original calculation from TEC data; solid curve is the smoothed results. The triangles, circles, and diamonds mark the time of the onsets, first emergences, and the peaks of the EIA crests during the evolution.

In the GPS TEC map (Figure 5a), the EIA northern crest appears earlier than the southern crest, at a local time near 0900. The emergence of the EIA has been marked as circles on the CTD (Figure 5c) curves as the CTD equals 0, which also exhibit northern crest priority. However, the local time of the marked emergence lags behind the crest's appearance on the TEC map (Figure 5a). This is because the calculation of CTD set the crest at fixed latitudes of 10°–15° N/S, while the real emergence of the EIA crest appeared at latitudes closer to the magnetic equator where the fountain effects launch. After the first emergence, CTD grows continuously, representing the development of EIA. The peaks of

the CTD mark that the EIA is fully developed and the TEC at the EIA crest reaches its peak value. Thus, we mark the peak CTD in the afternoon as the peaks of EIA.

Figure 5b,d show the IRI-2016 predicted TEC map and the corresponding CTD profiles, respectively. Besides a clearly different morphology of EIA structure (Figure 5b) compared with the GPS TEC measurements (Figure 5a), the IRI-2016 derived CTD also exhibited abnormal local time evolution. In detail (Figure 5d), there is no post-sunrise falling that ends in negative vales for NCTD; SCTD showed persistent positive value during 0700–2300 LT. Thus, the onset, as well as the first emergence of the EIA crest, can neither be identified on the IRI-2016 CTD. Only the peaks of the EIA crest can be marked as the peaks of CTD profiles. Note that a similar situation is a common feature for IRI-2016, though the three time points can be occasionally identified.

In summary, the local time of the onset, first emergence, and peak of EIA crest can be identified from GPS TEC measurement, while IRI-2016 cannot regularly capture the real time evolution process of the EIA development.

### 4.3. Time Evolution of EIA: Seasonal and Longitudinal Effects

In the last subsection, the time points of the onset, first emergence, and peak during the evolution of EIA can be marked on the CTD profiles. Figure 6 presents the variations of these time points as a function of months in 2013.

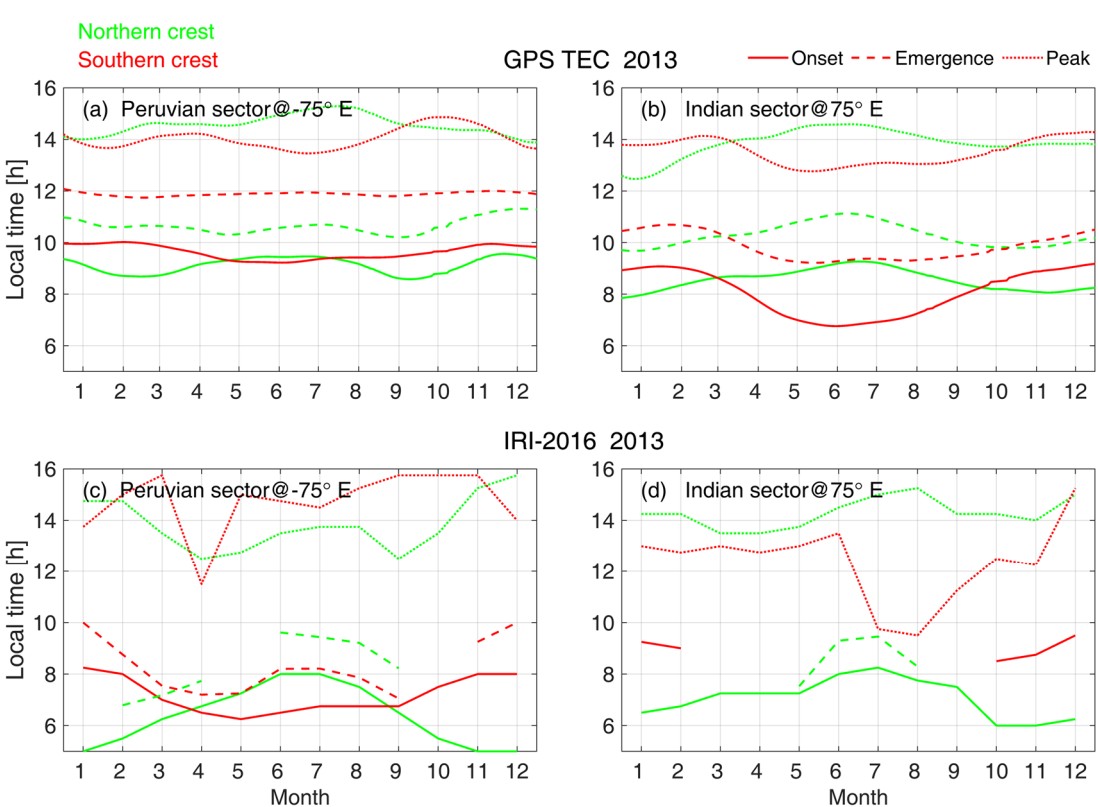

**Figure 6.** Local time of the onsets (solid lines), emergence (dash lines), peaks (dotted lines) of the EIA northern crest (green) and southern crest (red), at Peruvian (**a**–**c**) and Indian sector (**b**–**d**), extracted from GPS TEC (top panels) and IRI-2016 (bottom panels). The data gap in Figure 6c,d is due to that the time points sometimes cannot be identified in the IRI model.

Results from GPS TEC (Figure 6a,b) revealed that the patterned annual variations were generally shared by the three time points. The time lag between the first emergence of EIA crests with the onset is ~2 h, and it takes another ~4 h for the EIA crest to reach peak values. Specifically, semiannual and annual cycle patterns in the Peruvian and Indian sectors, respectively, can be captured. Take the onset of the northern EIA crest as an example, the

semi-annual cycle (Figure 6a) is characterized by an earlier onset time near two equinoxes and a later onset time near two solstices; the annual cycle (Figure 6b) is characterized by an earlier onset time in the winter and a later onset time in the summer. As for the southern crests, the time points generally exhibit reversed seasonal patterns compared with those of the northern crests, except that the emergence time at the Peruvian sector (Figure 6a) stays constantly near 1200 LT throughout the year 2013. Besides, the semiannual variation of the onset of the southern crest at the Peruvian sector is not as prominent as the northern crest, resulting in an earlier onset of southern crest than northern crest during winter seasons, which is similar to that at the Indian sector.

Figure 6c,d show the IRI-2016 predictions. As has been mentioned above, the onset and the emergence of EIA sometimes cannot be identified from IRI-2016 data. It can be witnessed that the onset and emergence data are sparse, due to the incapability for identifying those time points on CTD profiles. Nevertheless, the available time points generally predicted earlier onset and emergence, compared with those of the GPS observations. Interestingly, the annual cycle of EIA onset time is a prevailing phenomenon in both Peruvian and Indian sectors, while GPS revealed a semiannual cycle seen in the Peruvian sector. However, a similar semiannual cycle seen in GPS observation can be found on the peak time of the northern crest in the Peruvian sector (Figure 6c). Besides, the emerging time of the southern crest also exhibits semiannual variation (Figure 6c), but in the same phase of the time points of the northern crest seen in GPS observations (Figure 6a). It can be concluded that, although the semiannual cycle can be occasionally found on the emerging and peak time of EIA crest from IRI-2016, the onset time of EIA still shows the classic picture that the crest in the winter hemisphere develops earlier than that in the summer hemisphere. Note that the IRI-2016 derived peak time of EIA sometimes shows abnormal values (e.g., April at Peruvian sector, July and August at Indian sector, as shown in Figure 6c,d); this also indicated the inaccuracy of the IRI-2016 in retrieving EIA evolution.

The annual cycle shown in the Indian sector represents a classical picture of the neutral wind modulated ambipolar diffusion during the development of EIA. That is, the transequatorial wind blows from the summer hemisphere to the winter hemisphere, which pushes the plasma equatorward and poleward along the field line (referred to as pile-up effects) in the summer and winter hemispheres, respectively. These neutral wind effects contradict/favor the ambipolar diffusion in the summer/winter hemisphere; hence, the development of EIA crest in the summer/winter hemisphere is inhibited/promoted, resulting in that the time points of EIA's development show clear winter hemispheric priority.

Take the onset of the northern crest during the winter season as an example, the essential difference between the annual cycle (Indian sector, Figure 6b) and the semiannual cycle (Peruvian sector, Figure 6a) occurs during winter seasons. That is, the semiannual cycle consists of a delayed northern crest onset during northern winter (Figure 6a), which ought to occur earlier (Figure 6b) to exhibit the annual cycle. Thus, the fundamental question is what causes the retarded EIA northern crest development that is supposed to be advanced.

## 5. Discussion

The semi-annual cycle is a common feature on the overall intensity of plasma density at low and middle latitudes, which appears as density maximums at two equinoxes. The causes of the semi-annual cycle are discussed as follows. There is the ratio of atomic oxygen to molecular nitrogen $O/N_2$, which also shows similar semiannual variation, especially at middle latitudes [34]. In detail, during high $O/N_2$ ratio seasons (i.e., equinoxes), the chemical recombination rate is low, which causes a relatively high plasma density. He $O/N_2$ ratio is higher in the winter hemisphere, leading to a higher ionization rate [35]. However, this effect is more suitable to explain the middle latitudes winter anomaly [36] rather than that at low latitudes, since the hemispheric difference in the $O/N_2$ ratio is smaller in the low latitudes during the daytime. Thus, the $O/N_2$ ratio is less likely to contribute to the interhemispheric asymmetry of EIA's evolution seen in this study.

In addition to the $O/N_2$ ratio, the semi-annual cycle is also the property possessed by the daytime equatorial $E \times B$ drift that exhibits two equinoctial maximums and two solstitial minimums [37,38], via atmospheric tides modulated E-region dynamo [39]. Hence, the $E \times B$ drift is widely recognized as a major mechanism for the appearance of semiannual variation of the general EIA intensity [33]. However, there exist arguments regarding the $E \times B$ drift effects on the detailed EIA morphology. Wu et al. [40] attributed the semiannual variation of the northern EIA crest location to the $E \times B$ drift, while Liu et al. [14] doubt the mechanism as the daytime EEJ shows poor correlation with the EIA crest location.

The semiannual cycle of either the $O/N_2$ ratio or the $E \times B$ drift is a global phenomenon, if the $O/N_2$ ratio or the $E \times B$ drift is indeed involved in altering the efficiency of the EIA development, the earlier onset during equinoxes should also be witnessed in the southern hemisphere, and in other longitudes (i.e., Indian sector). However, this is not supported by the current observations. Hence, neither the $O/N_2$ ratio nor the $E \times B$ drift is responsible for the abnormal semiannual variation of the onset, emergence, and peaks of the EIA at the Peruvian sector, at least in a sole way.

The meridional neutral wind is recognized as the major impact of the asymmetric development of EIA. Thus, we raise a question of whether the semiannual cycle seen in the Peruvian sector is due to the local abnormal neutral wind configurations. To answer the question, the neutral wind result from TIEGCM simulation during two solstices and two equinoxes is adopted, during the geomagnetic quiet periods. Note that when discussing the pushing effects from neutral wind to plasma, the zonal wind contribution under the presence of magnetic declination should be considered. Hence, the effective magnetic meridional neutral wind velocity (U) could be written as:

$$U = U_\theta \cos D + U_\varphi \sin D \tag{3}$$

where $U_\theta$ (positive southward) and $U_\varphi$ (positive eastward) are the meridional and zonal wind velocities, and $D$ (positive eastward) is the declination angle [11]. Figure 7 presented the effective magnetic meridional wind as a function of local time and dip latitudes, in the Peruvian sector and Indian sector during March Equinox, June Solstice, September Equinox, and December Solstice. The dashed lines mark the dip equator; the two shaded bars in each plot mark the 'fixed crest' defined in this study at dip latitude of $10° \sim 15°$ S/N.

To illustrate the wind's effects on the ambipolar diffusion during the initial stage of the EIA development, we focus on regions of the equatorward side of the two 'fixed crests' (regions between the dip equator and the crests, $0° \sim 10°$ S/N in dip latitude). All three time points shown in Figure 6 are generally within 0600~1500 LT, so we focus on this local time bin.

In the Peruvian sector, the southward winds prevail through geomagnetic low latitude regions (i.e., regions between two shaded bars), except for the region near the northern crest (still in the geographic southern hemisphere) during December Solstice. For the southern crest, the southward wind is strongest/weakest at June Solstice and December Solstice, respectively. Thus, the promotion of the ambipolar diffusion to the southern crest is most/least prominent at June/December Solstice, which should result in the earliest/latest onset of EIA southern crest, consistent with the results shown in Figure 6a (red solid line). For the northern crest, the strongest southward wind occurs during June Solstice, while the northward wind appears during December Solstice. Thus, the ambipolar diffusion is inhibited/promoted during June/December Solstice that would result in an earlier/latest onset of northern EIA crest, which is inconsistent with the observed semiannual cycle (Figure 6a).

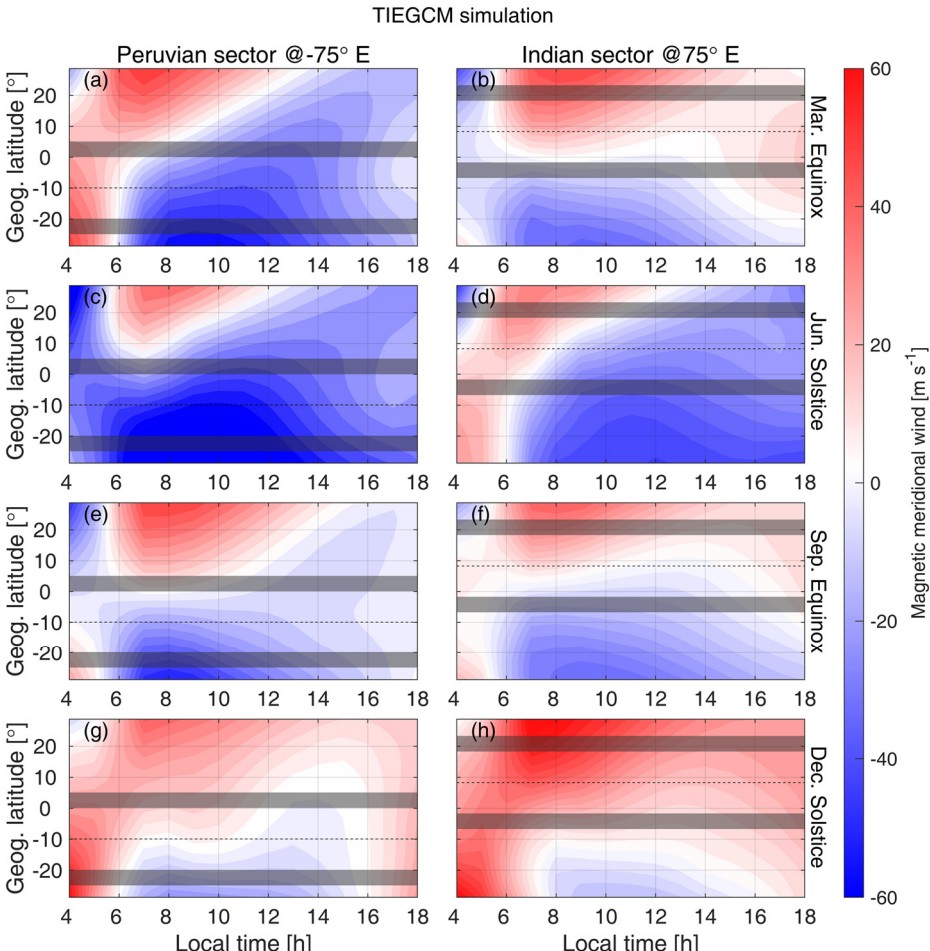

**Figure 7.** Effective magnetic meridional winds (positive northward). Avergaed winds centered on the days of (**a**,**b**) March Equinox, (**c**,**d**) June Solstice, (**e**,**f**) September Equinoxes, and (**g**,**h**) December Solstice, at Peruvian (left column) and Indian sector (right column). The dash line marks the dip equator, the two shaded bars in each plot mark the 'fixed crest' defined in this study at dip latitude of 10°~15° S/N.

In the Indian sector, the locations at both the northern and southern crests exhibit the strongest northward wind during December Solstice and the strongest southward wind during June Solstice. Hence, the ambipolar diffusion is severely retarded/promoted near southern/northern crest during December Solstice, and vice versa for June Solstice, resulting in an annual cycle on the onset of EIA, which is consistent with the observations (Figure 6b). We conclude that, under the assumption that the meridional wind only takes direct effects on the ambipolar diffusion, the wind configuration simulated by the TIEGCM would result in the typical annual cycle (winter crest priority) of the EIA evolution in both Peruvian and Indian sectors.

Note that the classical scenarios of the transequatorial wind effect on EIA have been challenged in both simulation [12,13] and observations [14]. Abdu [12] showed that the northward thermospheric wind would drive both the southern and northern EIA crest to move southward (upwind direction) in simulations, while the height of the F region is lifted/lowered near the northern/southern crest. Liu et al. [14] found an abnormal annual variation of EIA in the American sector; that is, the northern EIA crest resides at the highest/lowest latitude during local summer/winter when the southward/northward winds prevail, consistent with upwind seasonal movement of EIA crest proposed by Abdu [12], but contradicting the classical scenarios of the transequatorial wind effect.

So far, the physical mechanism of the tilt (in upwind direction) development of EIA has not been addressed either in the aforementioned simulation studies [12] or observational study [13]. Nevertheless, the mechanical effects of neutral wind would push the plasma not only equatorward/poleward but also upward/downward. The uplifting of the ionosphere leaves a depleted bottom side which encourages more ionizations; hence, the TEC should increase, and vice versa. In other words, though the equatorward wind retards the ambipolar diffusion, the accumulation of TEC near the EIA crest can still be accelerated by the uplifting of the ionosphere. In the same manner, the poleward wind that lowers the ionosphere would retard the speed of the growth of TEC near the EIA crest. Near the northern EIA crest at the Peruvian sector, the northward winds only occur at December Solstice (Figure 7h) when the F region resides at lower altitudes [41]. Thus, during the development of EIA, the TEC growth of the northern crest might be retarded, resulting in a delayed onset.

We emphasized that the above-proposed opposite neutral wind effects on the EIA evolution would depend on the longitude (magnetic declination) seasons, and possibly the solar activities. Hence, it is necessary to extend this study to a longer interval and more longitudes to further validate the proposed scenario.

## 6. Conclusions and Future Work Remarks

The study adopted the IGS TEC map and IRI-2016 to investigate the seasonal, interhemispheric variations on the time evolution (onset, first emergence, and peak) of EIA at Peruvian and Indian sectors. Major findings are listed below:

1.  Three time points can be concluded as: The onset occurs at 0600–1000 LT; the first emergence occurs at 0900–1200 LT; the peak occurs at 1200–1500 LT.
2.  The onset, first emergence, and peak of EIA show semiannual/annual cycle at the Peruvian/Indian sector. The annual cycle is characterized by a winter priority; that is, the EIA crest during local winter/summer develops earliest/latest. The semiannual is characterized as the northern/southern crest developing earlier during two equinoxes/solstices.
3.  The winter priority of the annual cycle can be explained by the transequatorial neutral wind that pushes the plasma along the field line to suppress/promote the EIA development in the summer/winter hemisphere. The semi-annual cycle might be associated with the effect of the neutral wind on the modulation of the F region height, which significantly alters the TEC.
4.  We suggest that the transequatorial wind would not only influence the EIA development via the modulation of ambipolar diffusion but also alter the F region height to further modulate the TEC growth speed. The two effects could be in a completive relationship, which causes complex seasonal variations of the EIA development. More studies are needed to further validate this mechanism.
5.  The IRI-2016 outputs generally underestimated the TEC value and showed abnormal interhemispheric asymmetry, and sometimes cannot correctly characterize the different stages of the EIA evolution, while the IGS TEC presented a more convincing pattern of the EIA evolution. We suggest that the lack of zonal electric field data that launches the ambipolar diffusion results in IRI's poor ability to describe the diurnal EIA evolution, and we indicate that the empirical model needs to be further improved. Thus, the I[2]RI-2016 model is not a good candidate to extend this study to longitudes where the GNSS observation is inadequate.

An interesting question remains: Why the F region height effect seems invalid in the Indian sector where the development of EIA showed typical winter crest priority. A possible explanation is that this competition varies at different longitudes/seasons/solar activity levels. In the future, by including the ionosonde data, we intend to extend this study to more longitude sectors and longer time intervals to further address the competition between the modulation of ambipolar diffusion and F region height variations.

**Author Contributions:** Conceptualization, X.W., C.X. and J.Z.; methodology, X.W. and J.Z.; software, X.W.; validation, X.W., H.W., C.X. and J.Z.; formal analysis, X.W.; investigation, X.W.; resources, J.C. and J.Z.; data curation, X.W.; writing—original draft preparation, X.W.; writing—review and editing, X.W., J.Z., C.X., H.W., Y.L., Q.L. and J.K.; visualization, X.W.; supervision, J.Z. and J.C.; project administration, X.W., J.Z., J.C. and J.K.; funding acquisition, X.W., J.Z., J.C. and J.K. All authors have read and agreed to the published version of the manuscript.

**Funding:** This research was funded by the Strategic Priority Research Program of the Chinese Academy of Sciences, grant number: XDB41000000; the National Natural Science Foundation of China, grant number: 41804150, 42104169, 42104147, 41804153, 41431073, 41521063, and 41674153; Guangdong Basic and Applied Basic Research Foundation, grant number: 2021A1515011216, 2020A1515110242; China Postdoctoral Science Foundation grant number: 2020M6830265, the Fundamental Research Funds for the Central Universities, Sun Yat-sen University, grant number: 2021qntd29; the Natural Science Fundation of Jiangsu Province, grant number: BK20180445; the Joint Open Fund of Mengcheng National Geophysical Observatory, grant number: MENGO-202018; the Opening Funding of the Chinese Academy of Sciences dedicated for the Chinese Meridian Project; the Open Research Project of Large Research Infrastructures of CAS—"Study on the interaction between low/mid-latitude atmosphere and ionosphere based on the Chinese Meridian Project".

**Data Availability Statement:** The International Global Navigation Satellite Systems Service (IGS) provide the GPS TEC map; data are available on the Coordinated Data Analysis Web (CDAWeb): Ftp://Cdaweb.Gsfc.Nasa.Gov/Pub/Data/Gps/Tec15min_Is/. The Kp index can be accessed on http://wdc.kugi.kyoto-u.ac.jp/kp/index.html#LIST.

**Acknowledgments:** Geomagnetic field measurements were collected at Fuquene, Huancayo, Tirunelveli, and Alibag. The author thanks T. A. Siddiqui for providing the geomagnetic field data from Huancayo and Fuquene; Virendra Yadav for providing the geomagnetic field data from Tirunelveli and Alibag. We thank the Colombian Instituto Geográfico Agustin Codazzi, the Instituto Geofisico del Peru, and the Indian Institute of Geomagnetism for supporting geomagnetic observatory operations. We also thank the World Data Center for Geomagnetism at Kyoto for providing the Kp index product.

**Conflicts of Interest:** The authors declare no conflict of interest.

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
