# Peer review of "Seasonal and Interhemispheric Effects on the Diurnal Evolution of EIA: Assessed by IGS TEC and IRI-2016 over Peruvian and Indian Sectors"

_remotesensing, doi:10.3390/rs14010107_

Round 1
Reviewer 1 Report
Report:
The work entitled "Seasonal and interhemispheric effects on the diurnal evolution of EIA: assessed by IGS TEC and IRI-2016 over Peruvian and Indian sectors" by Xin Wan, Jiahao Zhong , Chao Xiong, Hui Wang, Yiwen Liu, Qiaoling Li, Jiawei Kuai, Libin Weng and Jun Cui. It is a well-structured article that could be published after some minor observations and clarify some questions.
1.- Line 123: I suggest replacing the expression " that disturbed " by " that are distributed” .
2.- Lines 127: Replace the phrase: “which is not suitable for extending” by “, it is not suitable to extend”.
3.- Lines 248: I think that “(figure 2b)” does not correspond
4.- Line 400: Replace “Since the” by “The”…
- b) Questions:
1.- Figure 6: I believe that the interpolation performed in figures (6c) and (6d) do not correspond. In Figure (6c), in the month 3 to 5 interval, the interpolation is out of trend; similarly, in the month 6 to 10 interval, the interpolation shows a strange behavior.
2.- A general comment I think is necessary: The article is clear in analyzing the seasonal and interhemispheric effects on the diurnal evolution of the equatorial ionization anomaly (EIA). The authors use the (GNSS) Service (IGS), and the (IRI-2016) model. However, I don't know what sense it makes to have used the (IRI-2016) model if the results obtained are not good. It worries me when software is used as a black box. The question arises, what are the shortcomings of the IRI-2026 model? Why didn't they use the NeQuick model?
Reviewer 2 Report
The characterization of the equatorial electrojet (EEJ) and the clarification of the dynamical competition between the ionization of the sunlight and the resistance of the neural wind from the past (2013) is an interesting topic today when it comes to more intense events on the Sun. The research is interesting and promising, but the explanation is not very well done and should be presented more clearly. The problem I have is presenting the results and writing the text directly from the data sets to the results. The text and figures also seem out of sync. I wonder why the authors did not present the methodology of the research.
Therefore, I have some suggestions that the authors should consider/explain.
- Please introduce the reader to the research questions and the strategies used to answer them. The processing scenarios carried out in the manuscript are not clear or are too convoluted.
- In lines 126 and 127, the authors claim that: "Because there may be some inaccuracies in the data from IGS TEC in regions where original data from GPS are sparse, it is not possible to extend this study to these regions." Please cite previous research on this problem.
- Between section 2 and section 3, there is no explanation of the methodology. Please provide this as you cannot go directly from the data-sets to the results. Perhaps you should provide the reader with information about the geomagnetic rest conditions, calculation of Kp, etc.
- Lines from 156 to 172 should be included in the Methodology and Methods section.
- The equation in line 272 should be more visible (hence the suggestion not to include the equation in the text, but to put it in a separate place for equations).
- DOIs should be added in the references.
Again, the authors should reconsider the structure of the paper, because the topic is promising.
Reviewer 3 Report
The manuscript presents and discusses a vibrant topic of ionization modelling in the Peruvian and Indian sectors. The results of such studies are essential ex. for geodetic applications; hence, their interdisciplinarity aspect is convincingly exposed. The article has scientific soundness, and the extensive literature overview supports the modelling. Moreover, the visualization is pertinent and adequately performed. Nevertheless, while reading the material, I spotted some minor shortcomings needing improvements before publishing.
- The abstract should contain a slight justification for undertaking that particular theme. Also, the text lacks information about the uniqueness of the chosen regions. Some information is presented in lines 400-403, but the introductory section also needs clear motivation.
- Lines 380 - 387: such sections refer instead to the literature overview. I would recommend placing only comments about the results obtained in the discussion. Citing literature, referring to previous works fits better to sections 1 or 2.
- Lines 404-409: that information also refers to the literature overview. Here, only conclusions supported by research results should be presented.
- The section beginning in lines 419 and 431 partially answer the question of why the authors had chosen the discussed regions for their investigation. Nevertheless, I would suggest exposing it more clearly in the abstract and introduction.
- Line 449: choosing the specific reference with its results needs justification. Are there any other similar studies available?
- Conclusions are convincing and credible. I suggest referring to them while restructuring the article.
Should the authors perform necessary changes, the text can proceed with publishing. I congratulate the authors on their interesting studies.
Reviewer 4 Report
The paper presents results of the analysis of the GNSS TEC variations during 2013 observed at two near-equatorial locations (India, 75E and Peru, 75W). The development of the equatorial ionization anomaly (EIA) is studied during different seasons under geomagnetically quiet conditions. The observations are compared to the IRI-2016 simulations made for corresponding days.
Overall I have a good impression of the work done, and the paper is written in a clear and concise manner, however some questions arise.
The main concern is that the authors discuss the seasonal variations of different EIA parameters (the strength, onset, max etc. hours, hemispheric asymmetries, longitudinal differences) using the data of just one year. While this can be justified somehow for the equinoxes (the data for 4 months are used), the solstice features are obtained just for 1 month each. Authors should justify their choice of the studied time interval or extend their analysis to other years and show that the, e.g., longitudinal differences (annual/semiannual cycles) are not the features of that particular year.
Also, their arguments for the development of the semiannual cycle in the Peruvian sector are not convincing. Please add some calculations/plots to justify the mechanism you argue.
Other comments:
Lines 210-211: “solid black lines 210 represent 4-month averages”. I’m sure that not all means are calculated using 4 months data since both solstice plots contains data for 1 month only
Lines 277-308 and Fig.5: Why “typical” examples are shown and not the means? How are these “typical” examples chosen?
Round 2
Reviewer 2 Report
Dear Authors,
Thank you for the revisions and corrections to your manuscript. In its present form it is much more readable, even for beginners. I congratulate you on your work and your success.
The Reviewer from Slovenia
Reviewer 4 Report
I'm satisfied with the answers. The paper can be accepted